# CBAM: A Contextual Model for Network Anomaly Detection †

Henry Clausen [1,*] , Gudmund Grov [2,3] and David Aspinall [1,4]

1    School of Informatics, University of Edinburgh, Edinburgh EH8 9AB, UK; david.aspinall@ed.ac.uk
2    Norwegian Defence Research Establishment (FFI), 2007 Kjeller, Norway; Gudmund.Grov@ffi.no
3    Department of Informatics, University of Oslo, 0373 Oslo, Norway
4    The Alan Turing Institute, London NW1 2DB, UK
*    Correspondence: henry.clausen@ed.ac.uk
†    This paper is an extended version of our paper published in MLN'2020.

**Abstract:** Anomaly-based intrusion detection methods aim to combat the increasing rate of zero-day attacks, however, their success is currently restricted to the detection of high-volume attacks using aggregated traffic features. Recent evaluations show that the current anomaly-based network intrusion detection methods fail to reliably detect remote access attacks. These are smaller in volume and often only stand out when compared to their surroundings. Currently, anomaly methods try to detect access attack events mainly as point anomalies and neglect the context they appear in. We present and examine a contextual bidirectional anomaly model (CBAM) based on deep LSTM-networks that is specifically designed to detect such attacks as contextual network anomalies. The model efficiently learns short-term sequential patterns in network flows as conditional event probabilities. Access attacks frequently break these patterns when exploiting vulnerabilities, and can thus be detected as contextual anomalies. We evaluated CBAM on an assembly of three datasets that provide both representative network access attacks, real-life traffic over a long timespan, and traffic from a real-world red-team attack. We contend that this assembly is closer to a potential deployment environment than current NIDS benchmark datasets. We show that, by building a deep model, we are able to reduce the false positive rate to 0.16% while effectively detecting six out of seven access attacks, which is significantly lower than the operational range of other methods. We further demonstrate that short-term flow structures remain stable over long periods of time, making the CBAM robust against concept drift.

**Keywords:** network intrusion detection; deep learning; anomaly detection; flow prediction; access attacks

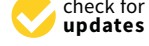



## 1. Introduction

Remote access attacks are used to gain control or access information on remote devices by exploiting vulnerabilities in network services and are involved in many of today's data breaches [1,2]. A recent survey [3] showed that these attacks are detected at significantly lower rates than more high-volume probing or DoS attacks. We present CBAM , a short-term Contextual Bidirectional Anomaly Model of network flows, which improves detection rates of remote access attacks significantly. The underlying idea of CBAM is to capture probability distributions over sequences of network flows that quantify their overall likelihood, much like a language model. CBAM is based on deep bidirectional LSTM networks. This paper is an extension of our previously presented work [4].

We evaluated CBAM carefully on three modern network intrusion detection datasets. By carefully selecting input parameters based on their sequential interdependence as well as increasing model complexity in terms of depth and efficient input embedding compared to preceding models, we are able to detect remote access attacks at a false positive rate of 0.16%, a rate at which none of the comparison models are able to detect any attacks reliably. CBAM is both able to detect six out of seven access attacks in the CICIDS-17 benchmark dataset and identifies traffic from real-world attacks in the LANL-15 dataset.

We also discuss specific design choices and how they enable the effective modelling of specific traffic characteristics to boost performance. Recently, deep learning models such as LSTMs have been a popular tool in network intrusion detection [5–7]. However, the evaluation of these models is generally agnostic to particular characteristics of the modelled traffic and fails to explain where and why the corresponding model fails to classify traffic properly. We recently demonstrated how a detailed examination of these failings of two state-of-the-art NID models enables specific improvements in the model design to boost detection performances [8]. The additional results presented herein aim to do the same for our short-term CBAM model and provide a validation for the undertaken design steps.

### 1.1. Contribution

This paper extends our work "Better Anomaly Detection for Access Attacks Using Deep Bidirectional LSTMs", presented at the Third International Conference on Machine Learning for Networking (MLN'2020) [4], and provides additional traffic examinations and corresponding performance results to demonstrate the design process. In particular, we present the following contributions:

- We provide extensive examination how common access attacks perturb short-term contextual sequence structures, and can thus be detected as anomalies by our presented model.
- We examine in detail how adding bidirectional LSTM-layers, increasing network depth and adding a separate size vocabulary helps CBAM predict flows better for specific traffic types to reduce false-positives.
- We provide additional performance results on the LANL-15 real-world dataset to demonstrate that CBAM is capable of detecting real-world attacks in real-world traffic.
- We examine in more detail how short-term flow sequence structures remain stable over long time periods, what the most common sources for false-positives are, and how the size of the training data affects the model's ability to reliably recognise benign traffic.

### 1.2. Outline

We now briefly explain the structure in which this paper is written, which is also visualised in Figure 1:

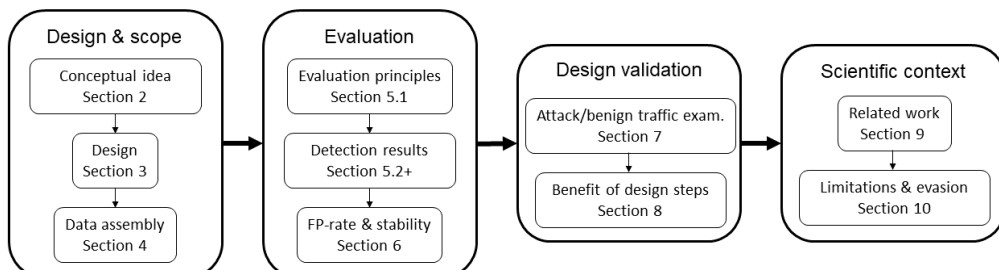

**Figure 1.** Structure of this paper.

**Design and scope:** Section 2 provides a motivation for short-term contextual models and their benefits for the detection of access attacks. Section 3 explains the methodology and architecture of CBAM as well as the data preprocessing. Section 4 describes the problems with network traffic datasets which previous methods were evaluated on and explains the advantages of our selection of datasets. We also describe how and why traffic from particular hosts is selected and how training and test data are constructed.

**Evaluation:** Section 5 discusses evaluation principles, our detection rates on attack traffic in the CICIDS-17 and LANL-15 data and examines how and why CBAM is able to identify these attacks. Section 6 discusses the false positive rate on benign traffic and examines the long-term stability of flow structures and corresponding model performance. It also provides details on the score distributions and the type of traffic that is both pre-

dicted accurately and inaccurately, as well as the influence of the training data size on
these predictions.

**Design validation:** Section 7 examines through given examples how attacks perturb
reoccurring traffic structures and how CBAM processes them. We also discuss how
different types of benign traffic are processed and the major cause for false-positives.
Section 8 discusses the reason and measured benefit of specific design steps that increase
model complexity.

**Scientific context:** Section 9 discusses related works and puts our method into context.
Section 10 highlights potential shortcomings and resilience against evasive tactics. Section 11
concludes our results.

## 2. CBAM: Flow Anomalies as Deviation from Predicted Sequences

In verbal or written speech, we expect the words "I will arrive by ..." to be followed
by a word from a smaller set such as "car" or "bike" or "5 p.m.". Similarly, on an average
machine, we may expect DNS lookups to be followed by outgoing HTTP/HTTPS connec-
tions. These short-term structures in network traffic are a reflection of the computational
order of information exchange. Attacks that exploit vulnerabilities in network communi-
cation protocols often achieve their target by deviating from the regular computational
exchange of a service, which should be reflected in the generated network pattern.

Table 1a depicts a flow sequence from an XSS attack. Initial larger flows are followed
by a long sequence of very small flows which are likely generated by the embedded attack
script trying to download multiple inaccessible locations. Flows of this size are normally
immediately followed by larger flows, as depicted in Figure 2, which makes the repeated
occurrence of small HTTP flows in this sequence very unusual.

**Table 1.** The left side depicts a flow sequence from an XSS attack. The right side depicts a benign
SMB-sequence (top) and a sequence from a *pass-the-hash* attack via the same SMB service.

| (a) XSS attack, A = 192.168.10.50, B = 172.16.0.1 | | | | |
|---|---|---|---|---|
| Src | Dst | DPort | bytes | # packets |
| A | B | 80 | 247,956 | 315 |
| A | B | 80 | 7544 | 13 |
| A | B | 80 | 328 | 6 |
| A | B | 80 | 2601 | 10 |
| A | B | 80 | 328 | 6 |
| A | B | 80 | 328 | 6 |
| A | B | 80 | 380 | 7 |
| A | B | 80 | 328 | 6 |
| | | ⋮ | | |

| (b) Benign SMB, C = C6267, D = C754 | | | | |
|---|---|---|---|---|
| Src | Dst | DPort | bytes | # packets |
| D | C | N33 | 600 | 5 |
| C | D | 445 | 77,934 | 1482 |
| D | C | N33 | 600 | 5 |
| C | D | 445 | 5202 | 10 |

| (c) *Pass-the-hash* attack via SMB | | | | |
|---|---|---|---|---|
| Src | Dst | DPort | bytes | # packets |
| C | D | 445 | 4,106,275 | 2830 |
| C | D | 445 | 358,305,611 | 242,847 |

Table 1b depicts a regular SMB service sequence while Table 1c depicts a *pass-the-hash*
attack via the same SMB service. As shown, the flows to port N33 necessary to trigger the

communication on the SMB port are missing while the second flow is significantly larger than any regular SMB flows due to it being misused for exfiltration purposes.

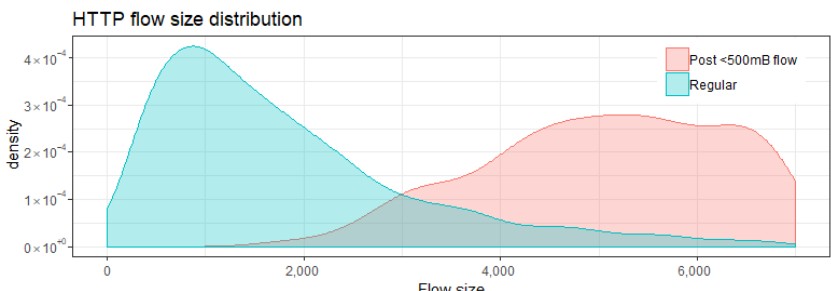

**Figure 2.** HTTP flow size distribution overall and if preceded by an HTTP flow smaller than 500 bytes.

The underlying idea of CBAM is to predict probabilities of connections in a host's traffic stream conditional on adjacent connections. The probabilities are assigned based on the connection's protocol, network port, direction and size and the model is trained to maximise the overall predicted probabilities.

To assign probabilities, we map each connection event to two discrete sets of states, called vocabularies, according to the protocol, the network port and the direction of the connection for the first, and according to the number of transmitted bytes for the second. The size of the vocabulary is chosen to be large enough to capture meaningful structures without capturing rare events that can deteriorate prediction quality. We feed these vocabularies into a deep bidirectional LSTM (long short-term memory) network that takes bivariate sequences of mapped events as input to efficiently capture the conditional probabilities for each event.

CBAM acts as an anomaly-detection model that learns short-term structures in benign traffic and identifies malicious sequences as deviations from these structures. By predicting probabilities of flows in benign flow sequences, CBAM is trained in a self-supervised way on strictly benign traffic. In contrast to classification-based training, CBAM does not require labelled attack traffic in the training data and is thus not affected by typical class imbalances in network intrusion datasets.

## 3. Design

### 3.1. Session Construction

A network flow (often referred to as "NetFlow") is a summary of the connection between two computers and contains a timestamp, the used IP protocol, the source and destination IP address and network port and a choice of summary values. The raw input data, in the form of network flows, contains unordered traffic from and to all hosts in the network. To order the raw network flows, we first gather all outgoing and incoming flows for each of the hosts selected for examination according to their IP address.

The traffic a host generates is often seen as a series of *sessions*, which are intervals of time during which the host is engaged in the same continued activity [9]. In our context, flows that occur during the same session can be seen as having strong short-term dependencies. We therefore group flows going from or to the same host to sessions using an established statistical approach [9]:

*If a network flow starts less than α seconds after the previous flow for that host, then it belongs to the same session; otherwise, a new session is started. If a session exceeds β events, a new session is started.*

We chose the number of $\alpha = 8$ seconds as we have found that on average, around 90% of flows on a host start less than 8 seconds after the previous flow, a suitable threshold to create cohesive sessions according to Rubin-Delanchy et al. [9]. We introduced the $\beta$ parameter in order to break up long sessions that potentially contain a small amount of

malicious flows and estimated $\beta = 25$ to be a suitable parameter. However, detection rates do not seem to be very sensitive to the exact choice of $\beta$.

A perfect session grouping would require (unavailable) information from the top layers of the network stack. We therefore use our session definition as a first approximation which we found to be useful enough for this experiment. We will discuss this issue further in Sections 8.2 and 10.

The interarrival time distribution for selected hosts in the used datasets, described in Section 4.1, along with 90% quantile lines, is depicted in Figure 3.

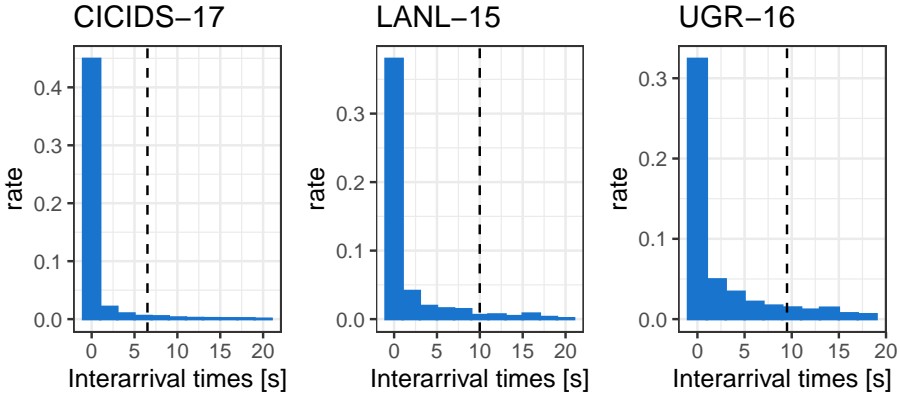

**Figure 3.** Flow interarrival distributions for selected hosts in the CICIDS-17, the LANL-15 and the UGR-16 data, with 90 percent quantile lines.

### 3.2. Contextual Modelling

Each session is now a sequence of flows that are assumed to be interdependent. We observed in an initial traffic analysis that the protocol, port and direction of a flow as well as its size are highly dependent on the surrounding flows, which motivates their use in the modelling process. We treat flows as symbolic events that can take on different states, much like words in a language model. The state of a flow is defined as the tuple consisting of the protocol, network port and the direction of the flow. We consider only the server port numbers, which indicate the used service, in the state-building process. We introduce the following notation:

| | |
|---|---|
| $M$: | number of states |
| $C$: | number of host groups |
| $S$: | number of size groups |
| $N_{\text{embed}}^i$: | embedding dimension |
| $N_{\text{hidden}}^i$: | LSTM layers dimension |
| $N_j$: | the length of session $j$ |
| $x^{i,j} \in \{1, \ldots, M\}$: | the state of flow $i$ in session $j$ |
| $c^j \in \{1, \ldots, C\}$: | the host group |
| $s^{i,j} \in \{1, \ldots, S\}$: | size group of flow $i$ in session $j$ |
| $p_x^{i,j,k} = P(x^{i,j} = k \mid j)$ | the predicted probability of $x^{i,j} = k$ conditional on the other flows in session $j$ |
| $p_s^{i,j,l} = P(s^{i,j} = l \mid j)$ | the predicted probability of $s^{i,j} = l$ conditional on the other flows in session $j$ |

The collection of all states is called a *vocabulary*. For prediction, the total size of a vocabulary directly correlates with the number of parameters needed to be inferred in an LSTM network, thus influencing the time and data volume needed for training. Too large vocabularies also lead to decreased predictive performance by including rare events that are hard to predict [10]. We therefore bound the total number of states and only distinguish between the $M - 2$ tuples of the protocol, port and direction most commonly seen on a machine, with less popular combinations being grouped as "other". Furthermore, the end of a session is treated as an additional artificial event with its own state. The total vocabulary size is then given by $M$.

Our experimentation has shown that detection rates improve when including the size as an additional variable, as we discuss in Section 7.1. Rather than making a point estimate of the size, we want to produce a probability distribution for different size intervals. This provides better accuracy for situations in which both small and large flows have a similar occurrence likelihood. We group flows into $S$ different size quantile intervals, with the set of all size intervals forming a third vocabulary. The $S - 1$ boundaries that separate the size intervals correspond to $S - 1$ equidistant quantiles of the size distribution in the training data.

Hosts are grouped according to their functionality (Windows, Ubuntu, servers, etc.), a distinction that can easily be performed using signals in the traffic. The group is provided to the model as an additional input parameter $c^j$ and forms a third vocabulary.

### 3.3. Architecture Selection

We now represent each session as a set of two symbolic sequences which contain between three and 27 items, in order to capture their contextual structure for the reasons described in Section 2. A number of techniques exist to describe such sequences, such as *Markov models and hidden Markov models, finite-state automata,* or *N-Gram* models. However, the success of recurrent neural networks in similar applications of natural language processing over these methods suggests they would be the most appropriate architecture to capture contextual relationships between flows. In Section 8.3, we compare the performance of CBAM to both Markov-based models and finite-state automata. Even though convolutional neural networks and feed-forward networks can be more suitable choices for specific sequential problems with tabular or regression characteristics, recurrent neural networks such as LSTMs or GRUs normally outperform them for short tokenised sequences [11]. Both LSTMs and GRUs perform similarly well and generally outperform simple RNNs.

### 3.4. Trained Architecture

We use a deep bidirectional LSTM network which processes a sequence in both forward and reverse direction to predict the state and size group of individual flows. The architecture of the network we trained is depicted in Figure 4. The increased model complexity we present has not been explored in previous LSTM applications to network intrusion detection and enables us to boost detection rates while lowering false positive rates, which we demonstrate in Section 8.

#### 3.4.1. Embedding

First, each of the three vectors is fed through an embedding layer, which assigns them a vector of size $N^i_{embed}$, $i \in \{1, 2, 3\}$. This embedding allows the network to project the data into a space with easier temporal dynamics. This step significantly extends existing designs of LSTM models for anomaly detection and allows us to project multiple input vocabularies simultaneously without a large increase in the model size. By treating the state, the size group and the host group as separate dictionaries, we avoid the creation of one large vocabulary of size $M \times C \times S$, which makes training faster and prevents the creation of rare states [10].

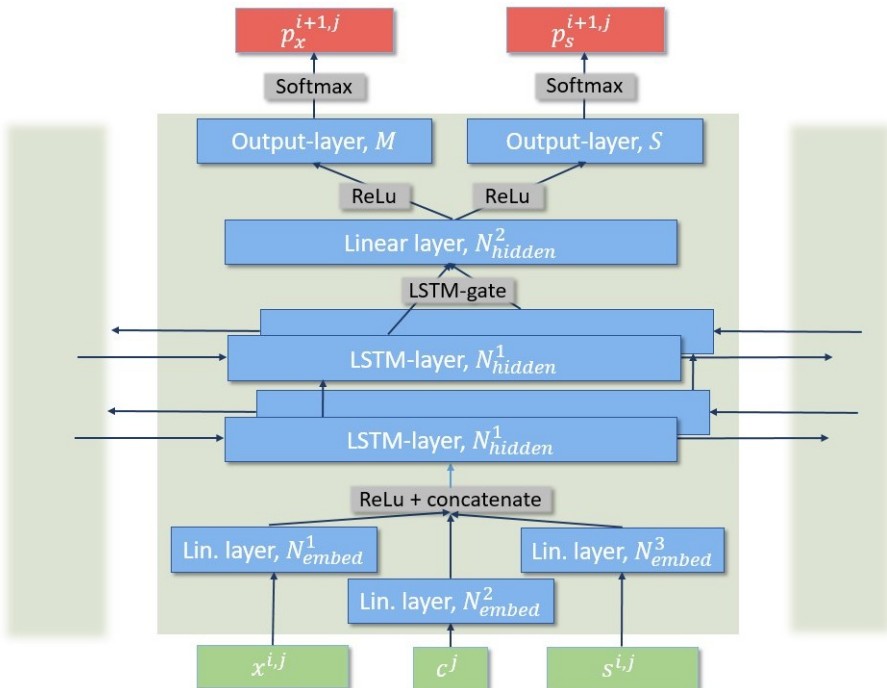

**Figure 4.** Architecture of the trained bidirectional LSTM network.

### 3.4.2. LSTM-Layer

In the second step, the vectors are concatenated and fed to a stacked bidirectional LSTM layer with $N^1_{\text{hidden}}$ hidden cells. This layer is responsible for the transport of sequential information in both directions. The usage of bidirectional LSTM layers compared to unidirectional ones significantly improved the prediction of events at the beginning of a session and consequently boosted detection rates within short sessions, as we demonstrate in Section 8.1. Increasing the number of LSTM layers from one to two decreases false positive rates in longer sessions while maintaining similar detection rates, as we show in Section 8.2. In Section 10.1, we discuss why we are not further increasing the number of layers.

### 3.4.3. Output Layer

The outputs from the bidirectional LSTM layers are now concatenated and fed to an additional linear hidden layer of size $N^2_{\text{hidden}}$ with the commonly used rectified linear activation function. We added this layer to enable the network to learn more non-linear dependencies in a sequence. We found that by adding this layer, we are able to capture complex and rare behaviours and decrease false positive rates, as demonstrated in Section 8.3.

Finally, the output of this layer is fed to two output layers with $M$ and $S$ softmax output cells. These produce two numeric vectors of size $M$ and $S$:

$$p_x^{i,j,k}, \ k \in \{1, \ldots, M\}, \qquad \sum_{k}^{M} p_x^{i,j,k} = 1$$

$$p_s^{i,j,l}, \ l \in \{1, \ldots, S\}, \qquad \sum_{l}^{S} p_s^{i,j,l} = 1$$

which describe the predicted probability distribution of $x^{i,j}$ and $s^{i,j}$, respectively.

The prediction loss for the state group is then given by the negative log-likelihood:

$$\text{lh}_x^{i,j} = \sum_{k=1}^{M} (1 - x_k^{i,j}) \cdot \log(1 - y_k^{i,j}) - x_k^{i,j} \cdot \log(y_k^{i,j})$$

with the size group loss being calculated in the same way. We calculate the total loss as the sum of the state loss and the size group loss. A visualisation of the prediction-making process is depicted in Figure 5.

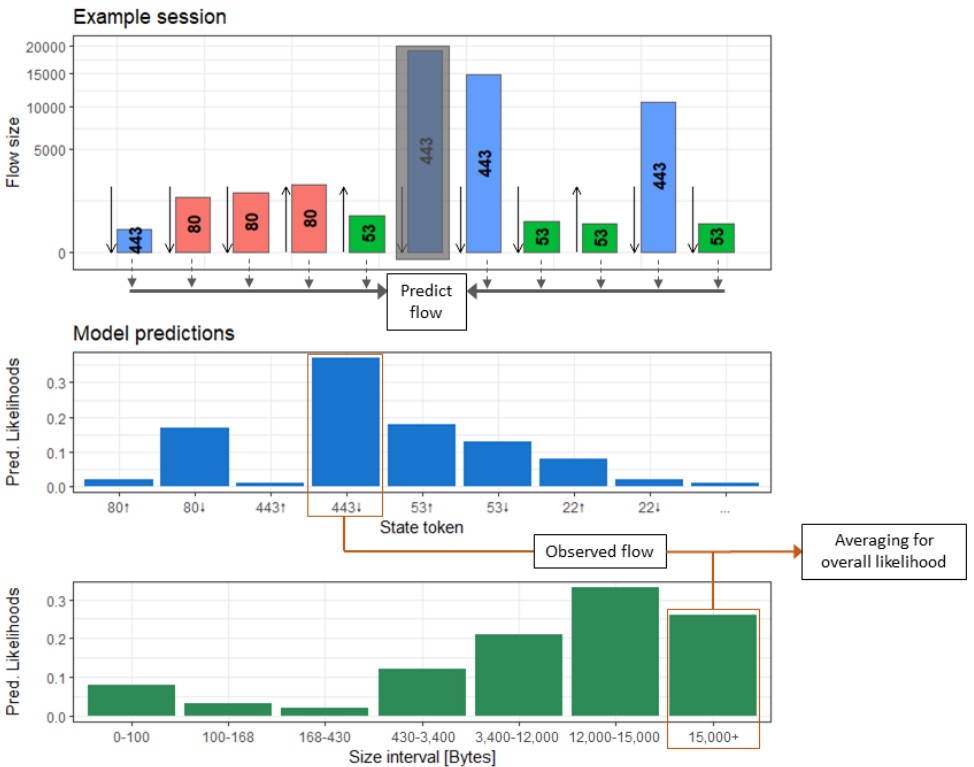

**Figure 5.** Visualisation of model prediction process.

After the training, we use the network to determine the anomaly score of a given input session via the average of the predicted likelihoods, as this measure is independent of the session length:

$$\text{AS}^j = 1 - \sum_{i=1}^{N_j} \left( \exp(\text{lh}_x^{i,j}) + \exp(\text{lh}_s^{i,j}) \right) / N_j$$

An anomaly score close to 0 corresponds to a benign session with a very high likelihood while a score close to 1 corresponds to an anomalous session with events which the network would not predict in the context of previous events. We rescaled all anomaly scores, however, this was done purely for better readability and does not influence the ordering and thus the detection process.

*3.5. Parameter Selection and Training*

We trained CBAM and tuned it to maximise its prediction performance. We trained it on a quad-core CPU with 3.2 GHz, 16 GB RAM and a single NVIDIA Tesla V100 GPU, and we used minibatches of size 30 using the ADAM optimiser in PyTorch. Training a model can be achieved in under three hours.

We aimed to create a model that has sufficient parameters to capture complex flow dependencies but does not overfit the training data. For this, we split the available training data into a larger training and a smaller validation set. We then selected two model configurations, one with a larger number of parameters and one with a smaller number. We then trained the model for 500 epochs on the training set and observed whether the same loss decrease can be observed on the validation set. As long as the larger model performed better than the smaller model and the validation loss was consistent with the training loss,

we kept increasing the number of parameters, a standard practice to train deep learning models. The best performing parameters were $N^1_{embed} = 10$, $N^2_{embed} = N^3_{embed} = 5$ for the embedding layers and $N^1_{hidden} = N^2_{hidden} = 50$ for the hidden layers.

To build a more powerful model without the risk of overfitting, we used a dropout rate of 0.5 as proposed by Hinton et al. [12], and a weight-decay regularization of $5 \times 10^{-4}$ per epoch. To increase the training performance, we used an adaptive learning of 0.0003, which decays by a factor of 2 after each fifty subsequent epochs, as well as layer normalization. The values for the learning rate and weight decay were estimated in a similar procedure as the model size.

As we mentioned above, too large vocabularies can cause problems both for model training and event prediction. We achieved the best results for $M = 200$ for the available data and computational resources. The size of the group was chosen to be smaller with $S = 7$, which improves the detection capabilities without increasing anomaly scores for benign sessions too much. We found that a suitable value of $C = 4$ can describe different host types, which include servers, two types of client machines depending on the operating system and auxiliary devices (printers, IP-phones and similar). Host groups were individually determined for each dataset and the corresponding machines were labelled manually. However, for larger datasets, this process is easily automated by filtering for specific traffic events such as requests for Microsoft update servers.

|  | # Cells | # Parameters |
|---|---|---|
| Embedding layer | 202/10/5 | 2055 |
| LSTM-layer 1 | 50 | 6,700 |
| LSTM-layer 2 | 50 | 12,700 |
| Linear layer | 50 | 2550 |
| Softmax layer | 202/10 | 10,557 |
| Total | 34,562 | |

### 3.6. Detection Method

We used a simple threshold anomaly score to identify a session as malicious. We estimated the 99.9% quantile for benign sessions in the training data, which will then act as our threshold value T. By determining T from the training data, we controlled the expected false positive rate in the test data. Threshold values were determined for each dataset and each host within a dataset separately:

$$T_c : P[AS^j_{cj} \leq T_c] \leq 0.999$$

Finding an appropriate threshold value is a compromise between higher detection rates and lower false positive rates, and we chose this value to achieve false positive rates that are low enough for a realistic setting. We compared the detection and false positive rates for a different T in Section 5.2, and we give an outlook to more sophisticated detection methods in Section 10.

## 4. Datasets

### 4.1. Dataset Assembly

The field of network intrusion detection has always suffered from a lack of suitable datasets for evaluation. Privacy concerns and the difficulty of posterior attack traffic identification are the reason that no dataset exists that contains realistic U2R/R2L (user-to-root, remote-to-local) traffic and benign traffic from a real-world environment [13]. To evaluate CBAM, we need both representative access attack traffic to test detection rates, and background traffic from a realistic environment to test false positive rates. To ensure that both criteria are met, we selected three modern publicly available datasets that complement each other: CICIDS-17 [14]; LANL-15 [15]; and UGR-16 [16]. The CICIDS-17 dataset contains traffic from a variety of modern attacks, while the UGR-16 dataset's length is

suitable for long-term evaluation. The LANL-15 dataset contains enterprise network traffic along with several real-world access attacks.

We trained models with the same hyperparameters on each dataset to demonstrate the capability of CBAM to detect various attacks and perform well in a realistic environment.

**CICIDS-17:** This dataset [14], released by the Canadian Institute for Cybersecurity (CIC), contains 5 days of network traffic collected from 12 computers with attacks that were conducted in a laboratory setting. The computers all have different operating systems to enable a wider range of attack scenarios. The attack data of this dataset are among the most diverse among NID datasets and contain SQL injections, heartbleed attacks, bruteforcing, various download infiltrations, and cross-site scripting (XSS) attacks, on which we evaluated our detection rates.

The traffic data consist of labelled benign and attack flow events with 85 summary features which can be computed by common routers. The availability of these features makes it suitable to evaluate machine-learning techniques that were only tested on the KDD-99 data.

The benign traffic is generated on hosts using previously gathered and implemented traffic profiles to make the traffic more heterogeneous during a comparably short time span, and consequently closer to reality. For our evaluation, we selected four hosts that are subject to U2R and R2L attacks, two web servers and two personal computers.

This dataset is generated in a laboratory environment, with a higher proportion of attack traffic than is normally encountered in a realistic setting. Consequently, we need to test it on traffic from real-world environments to prove that CBAM retains its detection capabilities and low false alert rates.

**LANL-15 dataset:** In 2015, the Los Alamos National Laboratory (LANL) released a large dataset containing internal network flows (among other data) from their corporate computer network. The netflow data were gathered over a period of 27 days with approximately 600 million events per day [15].

In addition to large amounts of real-world benign traffic, the dataset contains a set of attack events that were conducted by an authorised red team and are supposed to resemble remote access attacks, mainly using the *pass-the-hash* exploit. We selected this dataset to demonstrate that CBAM is able to detect attacks in a realistic environment with low false alert rates. We isolated traffic from ten hosts, with two being subject to attack events. Two of these hosts resemble server behaviour, while the other eight show the typical behaviour of personal computers.

The provided red team events are not part of the network flow data and only contain information about the time of the attack and the attacked computer. Furthermore, not all of the attack events are conducted on the network level, so it is impossible to tell exactly which flows correspond to malicious activity and which do not. Therefore, we labelled all flows in a narrow time interval around each of the attack timestamps as possibly malicious. As these intervals are narrow, identified anomalies likely correspond to the conducted attack.

**UGR-16 dataset:** The UGR-16 dataset [16] was released by the University of Grenada in 2016 and contains network flows from a Spanish ISP. It contains both clients' access to the Internet and traffic from servers hosting a number of services. The data thus contain a wide variety of real-world traffic patterns, unlike other available datasets. Additionally, a main focus in the creation of the data was the consideration of long-term traffic evolution, which allows us to make statements about the robustness of CBAM to concept drift over the 163 day span of the dataset. For our evaluation, we isolated traffic from five web-servers that provide a variety of services.

Other Datasets

Two datasets and their derivatives, DARPA-98 [17] and KDD-99 [18], are often used to benchmark detection models, with all anomaly-based techniques discussed in a recent survey [3] with reported detection rates on U2R and R2L attacks relying on either of them. Both datasets have been pointed out as flawed and can give overoptimistic results due to

inconsistencies, a lack of realistic benign traffic, and an imbalance of benign and attack traffic [19–21]. Both datasets are 20 years old and outdated. The KDD-99 dataset was collected using a Solaris operating system in a laboratory environment to collect a wide range traffic and OS features, which makes it the most popular dataset to evaluate machine-learning-based techniques. The collection of many of these features is, however, currently infeasible in real-world deployment. All these factors mean that reported detection rates collected on these datasets have to be taken with care.

CTU 2013: The Stratosphere Laboratory [22] in Prague released this dataset in 2013 to study botnet detection. It consists of more than 10 million labelled network flows captured on lab machines for 13 different botnet attack scenarios. A criticism of this dataset is the unrealistically high amount of malicious traffic contained in the dataset, which makes it easier to spot it while reducing false positives. Furthermore, the way normal or background traffic is generated is described only poorly and leaves the question of how representative it is of actual network traffic.

UNSW-NB 2015: The dataset released by the *University of New South Wales* in 2015 [23] contains real background traffic and synthetic attack traffic collected at the "Cyber Range Lab of the Australian Centre for Cyber Security". The data were collected from a small number of computers which generate background traffic, which is overlayed with attack traffic using the *IXIA PerfectStorm tool*. The time span of the collection is in total 31 h. An advantage of the data is the variety of the attack data, which contain a number of DoS, reconnaissance and access attacks. However, due to the synthetic injection of these attacks, it is unclear how close they are to real-world attack scenarios, and again, the generation process of benign traffic is poorly described and leaves the question how close to the actual network traffic it is.

ADFA 2013/2014 [24]: The ADFA dataset, released by the *University of New South Wales*, focuses on attack scenarios on Linux and Windows systems as well as stealth attacks. To create host targets, the authors installed web servers and database servers, which were then subject to a number of attacks. The dataset is more directed towards attack scenario analysis and is criticised as being unsuitable for intrusion detection due to its lack of traffic diversity. Furthermore, the attack traffic is not well separated from the normal one.

CICIDS 2018 [14]: This dataset, released by the *Canadian Institute for Cybersecurity* (CIC), is generated in a similar fashion to the CICIDS 2017 data that we used in this work. The main differences are that the CICIDS 2018 data spans over three weeks and include in total 450 hosts but lacks the amount of web-attacks that we require and which are present in the CICIDS-2017 dataset.

The LITNET-2020 [25] dataset from the Kaunas University of Technology Lithuania from 2020 was collected from an academic network over a timespan of 10 months and contains annotated real-life benign and attack traffic. The corresponding network provides a large network topology with more than a million IP-addresses, and the data were collected in the form of network flows with more than 80 features. However, the dataset only contains traffic from high volume attacks such as DoS-, scanning, or worm attacks, which are not suitable to evaluate CBAM.

The Boğaziçi University distributed denial of service dataset [26] contains both benign traffic from more than 400 users from an academic network as well as artificially created DoS-attack traffic. The dataset spans only 8 min and contains no access attacks.

*4.2. Dataset Split*

We split our data into a test set and a training set. To resemble a realistic scenario, the sessions in the training data are from a previous time interval than sessions in the test data.

To evaluate detection rates on the CICIDS-17 data, we selected the four hosts in the data that are subject to remote access attacks, two web servers and two personal computers. We chose our test set to contain the known attack data while the training data should only contain the benign data. Due to the short timespan of the dataset, we had to train on traffic from all five days, with the test data intervals being placed around the attack. In total, the

test set contains 14 h of traffic for each host while the training set contains 31 h of traffic. While the test set for the CICIDS-17 data covers a shorter timespan, it contains more traffic due to voluminous brute-force attacks.

For the LANL data, the test set stretches approximately over the first 13 days with the training data spanning over the last 14 days. The unusual choice of placing the test set before the training set was made because the attacks occur early in the dataset. However, as the training and test are contained in two non-overlapping intervals, a robustness evaluation is still possible.

To test the long-term stability and robustness of CBAM against concept drift, we split the UGR-16 data into one training set interval and two test set intervals, for which we can compare model performance. The training set interval stretches over the first month, with the first test set interval containing the sessions from the following two months, and the second test set interval containing the last two months. We then isolated traffic from five web-servers that provide a variety of services that show behavioural evolution. Figure 6 depicts the changes of these servers in terms of protocol and port usage over the different intervals.

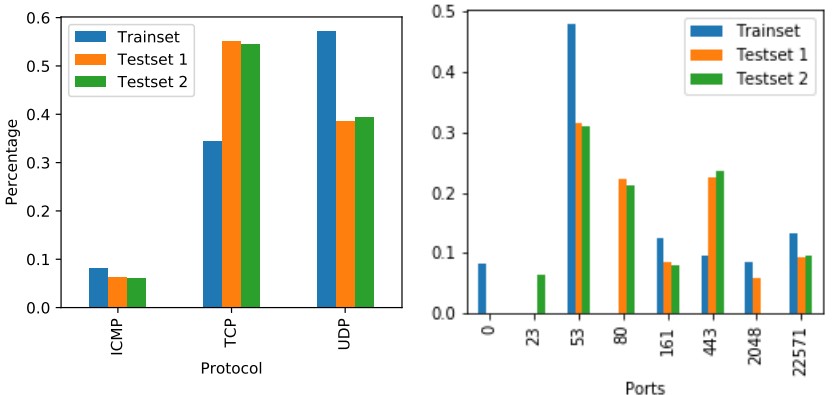

**Figure 6.** Temporal change in the protocol and port usage over the different train and test intervals across selected servers in the UGR-16 dataset.

We chose our training data to contain approximately 10,000 sessions per host if possible. A summary of the amount of data in the training and test data for each dataset can be found in Table 2.

**Table 2.** Summary of the amount of traffic extracted from each dataset.

| Dataset | Hosts | Sessions in Training Set | Sessions in Test Set | Length |
|---------|-------|--------------------------|----------------------|--------|
| CICIDS-17 | 4 | 24,128 | 32,414 | 5 days |
| LANL-15 | 10 | 89,480 | 76,984 | 27 days |
| UGR-16 | 5 | 65,000 | 480,018 | 163 days |

## 5. Detection Performance

### 5.1. Sample Imbalance and Evaluation Methodology

Most NID datasets include attack events from both low volume access attack classes as well as attacks like DoS or port scans which generate a large number of events. If reported detection rates do not distinguish between different attacks or attack classes, performance metrics will be dominated and potentially inflated by DoS and probing attacks. Similarly, detection results are often given in terms of precision and recall or F-measures, which are sensitive to the specific dataset balance of the majority and minority classes in a dataset. Since the ratio of attack traffic is inflated in general NID-datasets, these measures are not suitable for model comparison.

We evaluated the CBAM using simple true positive and false positive rates, which are independent of the dataset balance. We also distinguished detection rates for different attacks and assess overall performance by averaging these rates over attack classes rather than an overall number of attack events. Since there is no agreed upon value for a suitable false positive rate in network intrusion detection, we computed ROC-curves to display the detection rates in dependence of the false-positive rate and reported the overall AUC-scores (*area under curve*), which describe the separation of benign and anomalous traffic. We use these for comparison with other models, as this measure is fairer than point comparisons. The evaluation procedure is supported by several NID evaluation surveys [27,28].

Some researchers have proposed cost-based evaluation metrics by assigning false alerts and missed intrusion attempts a cost-value and tuning the detection threshold to minimise the expected cost, such as done by Ulvila and Gaffney [27]. Such a metric is, however, strongly dependent on the observed ratio of attack to benign traffic, which is strongly inflated in NID-datasets, and requires operational information to assign costs to a false alert or intrusion. This evaluation works well in specific cases such as DoS attacks or cryptojacking, where server-downtime costs and attack volume are generally quantifiable, but it is not applicable in cases where this information is not available or well defined [28].

Training classifiers on imbalanced datasets can affect their performance, both due to the imbalanced ratio of attack to benign traffic and the imbalance between several attack classes. Some methods have been proposed to augment or synthetically inflate minority samples for attack traffic [29]. As an anomaly-detection method, CBAM is, however, trained on a self-supervised way strictly on benign traffic, with no attack traffic being present in the training data. The training stage is therefore independent of the minority class ratio in a given dataset and does not require specific balancing methods. In the evaluation stage, the above-described steps apply for both classification- and anomaly-detection-based methods to consider data imbalances.

*5.2. CICIDS-17 Results*

We now demonstrate that we can build an accurate and close-fitting model of normal behaviour with CBAM. We train models for each dataset separately, but without any change in the selected hyperparameters, i.e., the number of hidden cells, vocabulary size, learning rate etc.

As described above, we estimated the detection rates using the traffic of various remote access attacks in the CICIDS-17 dataset. Table 3 describes the number of sessions present for each attack class.

**Table 3.** Number of sessions for each attack class in the CICIDS-17 dataset.

|            | FTP-BF | SSH-BF | Web-BF | SQL-Inj. | XSS | Heartbleed | Infiltr. |
|------------|--------|--------|--------|----------|-----|------------|----------|
| # Sessions | 243    | 210    | 88     | 8        | 41  | 4          | 17       |

Table 4 depicts anomaly score distributions and detection rates for traffic from seven different types of attacks.

Most notable is that scores from all attacks except cross-site scripting (XSS) are significantly higher distributed than benign traffic, with median scores lying between 0.75 and 0.89. Detection rates with our chosen threshold of $T = 0.77$ are highest for Heartbleed attacks (100%), followed by FTP and SSH brute-force attacks and SQL-injections, where 91%, 74% and 75% of all affected sessions are detected. Detection rates are lowest for XSS and infiltration attacks. The overall detection rates we achieve are in a similar range as most unsupervised methods in Nisioti et al.'s evaluation [3], but with significantly better false positive rates.

**Table 4.** Anomaly score distributions and detection rates at threshold T for known malicious sessions in the CICIDS-17 dataset, as well as detection rates for a less complex benchmark model described in Section 8.3.

| | Anomaly Scores (T = 0.77) | | | Detection | Shallow |
| | min | max | Median | Rates [%] | LSTM |
|---|---|---|---|---|---|
| Brute-force Web | 0.50 | 0.92 | 0.80 | 0.66 | 0.28 |
| FTP-Patator | 0.28 | 1.00 | 0.82 | 0.91 | 0.38 |
| Heartbleed | 0.89 | 0.89 | 0.89 | 1.00 | 0.0 |
| Infiltration | 0.57 | 0.97 | 0.75 | 0.41 | 0.0 |
| SQL-injection | 0.67 | 1.00 | 0.84 | 0.75 2 | 0.21 |
| SSH-Patator | 0.47 | 0.86 | 0.80 | 0.74 | 0.67 |
| XSS | 0.06 | 0.75 | 0.20 | 0.00 | 0.0 |

XSS and infiltration attacks cause the victim to execute malicious code locally. Heartbleed and SQL injections, on the other hand, exploit vulnerabilities in the communication protocol to exfiltrate information, and are thus more likely to exhibit unusual traffic patterns, visible as excessively long SQL-connections or completely isolated TCP-80 flows for SQL-attacks, or unusual sequences of connections initiated by the attacked server during heartbleed attacks.

Brute-force attacks on the other hand cause longer sequences of incoming connections to the same port of a server, in this case to port 21 for FTP, 22 for SSH, and 80 for web brute-force. Especially for port 80, such sequences are not necessarily unusual, which explains the difference in detection rates between web brute-force, which CBAM does not detect reliably, and FTP and SSH brute-force, which are detected at a higher rate. Depending on how much benign traffic the particular sessions are overlayed, the estimated anomaly scores can vary. Brute-force attacks are not low in volume and spread over many sessions since we introduced a maximum session length. For these types of attack, CBAM therefore only has to flag a smaller percentage of malicious sessions than the attack generates to detect anomalous behaviour.

Figure 7 provides *ROC* (receiver operating characteristic) curves for each attack type. As seen, for heartbleed, FTP brute-force, SQL injection, and infiltration attacks, CBAM starts detecting attacks with close to zero false positives.

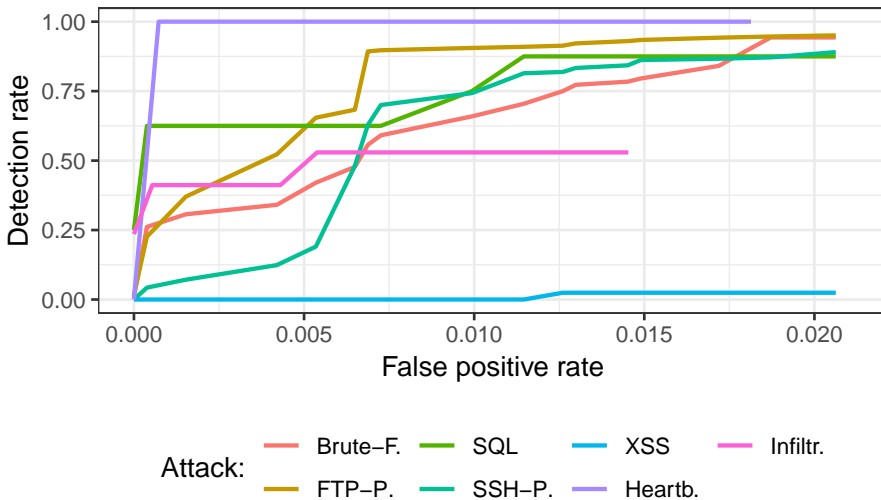

**Figure 7.** ROC curves for different attack types in the CICIDS-17 dataset.

*5.3. LANL Results*

We now examine whether CBAM is able to detect actual attacks in real-life traffic from the LANL-15 dataset.

As described in Section 4.1, we did not have labels for malicious flows in the LANL-15 data. Instead, attacks are described by narrow intervals surrounding conducted malicious activity. These intervals inevitably contain benign activity too. However, as the intervals are narrow and we saw that benign sessions only rarely receive high anomaly scores, a session with a high anomaly score is likely to be associated with a malicious event. Of the hosts in the dataset we selected for evaluation, hosts C2519 and C754 are subject to the red team attacks. The red team activity is spread over three attack intervals **A1**, **A2**, and **A3**.

Sessions in **A1** and **A2** have similar scores as other benign traffic, with no sessions receiving remarkably high scores. It is both possible that CBAM did not identify the malicious traffic, or that the activity in these intervals was purely host-based and did not generate any traffic.

Interval **A3** is more interesting, containing 15 sessions for host C2519 and five sessions for host C754 that have high anomaly scores, as depicted in Figure 8.

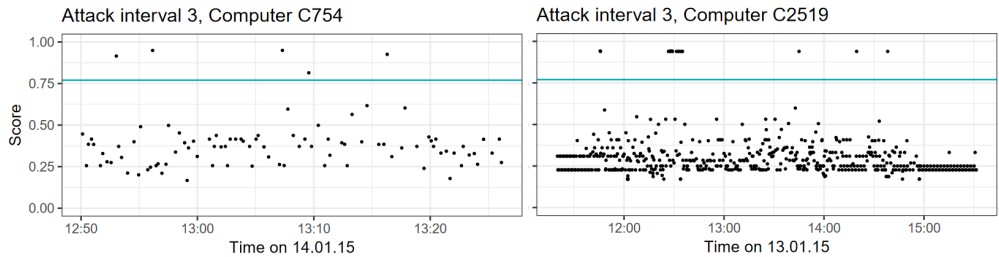

**Figure 8.** Computed scores for the third attack interval in the LANL data, along with our detection threshold.

For host C2519, each session with a high anomaly score consists of a single TCP-flow on port 445, which is usually reserved for a Microsoft SMB service. The anomaly of these sessions becomes apparent when we compare them to other sessions that contain TCP-flows on port 445, as depicted in Table 5.

**Table 5.** Exemplary session with SMB-traffic, host C2519, along with the estimated probability of the session to end.

|   | Proto | SrcAddr | Sport | DstAddr | Dport | Prob_End |
|---|-------|---------|-------|---------|-------|----------|
| 1 | tcp | C20473 | N345 | C2519 | 445 | 0.03 |
| 2 | tcp | C2519 | 445 | C20473 | N345 | 0.55 |

All other sessions contain at least two subsequent flows. The model, in expectation of other following flows, assigns a very low probability to the sessions ending after a single flow. Since the analysis of the identified sessions supports their anomaly score, we believe it is very likely that these events correspond to the conducted malicious activity.

### 5.4. Runtime Performance

CBAM contains around 35,000 parameters, which is relatively lightweight for deep learning models. The processing of a session of ten flows takes around 23 ms on our setup, which is far shorter than the average length of 5.6 s of a session. In a similar comparison, our setup can process one day of activity ($\approx$15,000 sessions) of a web server in the UGR-16 dataset in 95 s.

Considering these runtime numbers, the necessary rate of recorded flows to overwhelm our setup would need to exceed 434 flows/second. The largest rate observed for brute-force attacks in the CICIDS-17 dataset is 23 flows/second.

## 6. Benign Traffic and Longterm Stability

### 6.1. UGR-16 Data

We conducted the main validation of the long-term stability of CBAM on benign traffic in the UGR-16 dataset, which contains real-world traffic and spans several months. For this, we split the test data into two disjoint sets that span May–July and August–September while being separated by one month. We then look at the quantiles and visual distribution of session scores in each test set and assess whether the score distributions and number of false positives changed as evidence on concept drift in the traffic. Figure 9 depicts the score distribution of benign sessions for each dataset in the corresponding test sets.

As visible in the plot, the centre of the score distribution is concentrated very well in the lower region of the $[0, 1]$ interval, with about 50% of all sessions receiving scores in the region between 0.1 and 0.25. High scores are rare, with only very small percentages exceeding our chosen detection threshold of $T = 0.76$.

This is also reflected by the corresponding table that describes score distributions for all five hosts in the UGR-16 data. On average, less than 0.15% of all assumed-benign sessions exceed the threshold, which would translate to fewer than ten false-alerts over the span of four months on a host with similar activity rates.

Differences in the score distributions for the two test sets are quasi non-existent. The core of the distributions are very stable, with the score quantiles differences being less than 0.03. There are some differences in the observed false positives, but the available sample size is not large enough to make any statements on any systematic differences.

A clear banding structure is visible in the plotted distributions, with most session scores being very concentrated on narrow intervals. These scores represent frequently reoccurring activities that generate very similar traffic sequences. Figure 9 shows that these banding structures remain virtually unchanged over several months and carry over from test set 1 to test set 2.

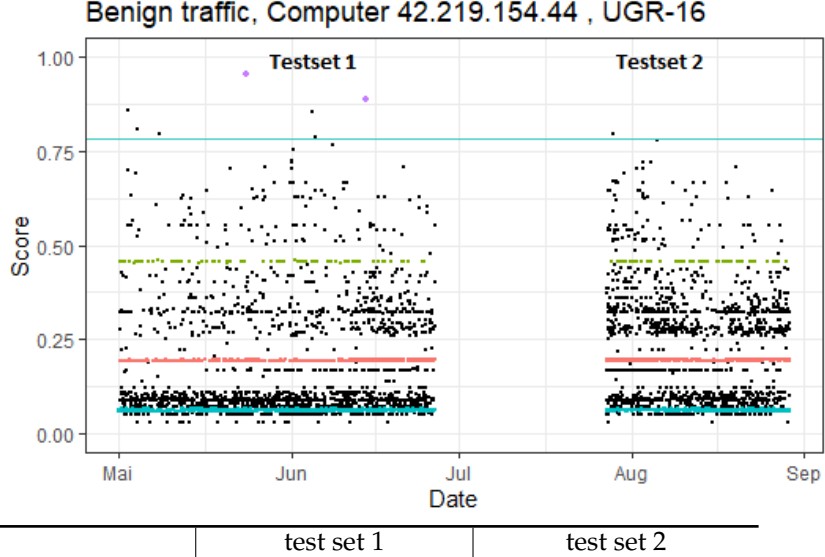

| | test set 1 | | | test set 2 | | |
|---|---|---|---|---|---|---|
| | 50% | 90% | Pr($>$T) | 50% | 90% | Pr($>$T)) |
| 42.219.153.32 | 0.21 | 0.39 | 0.01% | 0.22 | 0.38 | 0.01% |
| 42.219.155.189 | 0.12 | 0.20 | 0.01% | 0.10 | 0.21 | 0.03% |
| 42.219.155.128 | 0.24 | 0.44 | 0.63% | 0.19 | 0.43 | 0.41% |
| 42.219.155.4 | 0.13 | 0.34 | 0.10% | 0.17 | 0.39 | 0.23% |
| 42.219.154.44 | 0.11 | 0.32 | 0.13% | 0.11 | 0.28 | 0.11% |

**Figure 9.** Anomaly score distributions for benign traffic in the UGR-16 data, along with an exemplary distribution plot for a selected host.

### 6.2. CICIDS-17 and LANL-15 Results

We now look at the structure and stability of anomaly scores for benign traffic in the LANL-15 and CICIDS-17 datasets. The plots and tables in Figure 10 depict the score distribution of presumably benign sessions in both datasets as well as describe the 50% and 90% quantiles and false-positive rates for each host. Again, score distributions for both datasets are well concentrated in the lower region of the [0, 1] interval. For both datasets, the median lies between 0.06 and and 0.29.

For the LANL-15 data, we observe the same banding structure as in the UGR-15 data, with most sessions being concentrated in these bands. This banding is, however, far less pronounced in the CICIDS-17 data, with the majority of session scores here being dispersed to a greater extent. This suggests that the traffic generation process for this dataset relies far less on reoccurring rigid activities than we observe in real-life data, which, however, does not seem to deteriorate the prediction performance of CBAM.

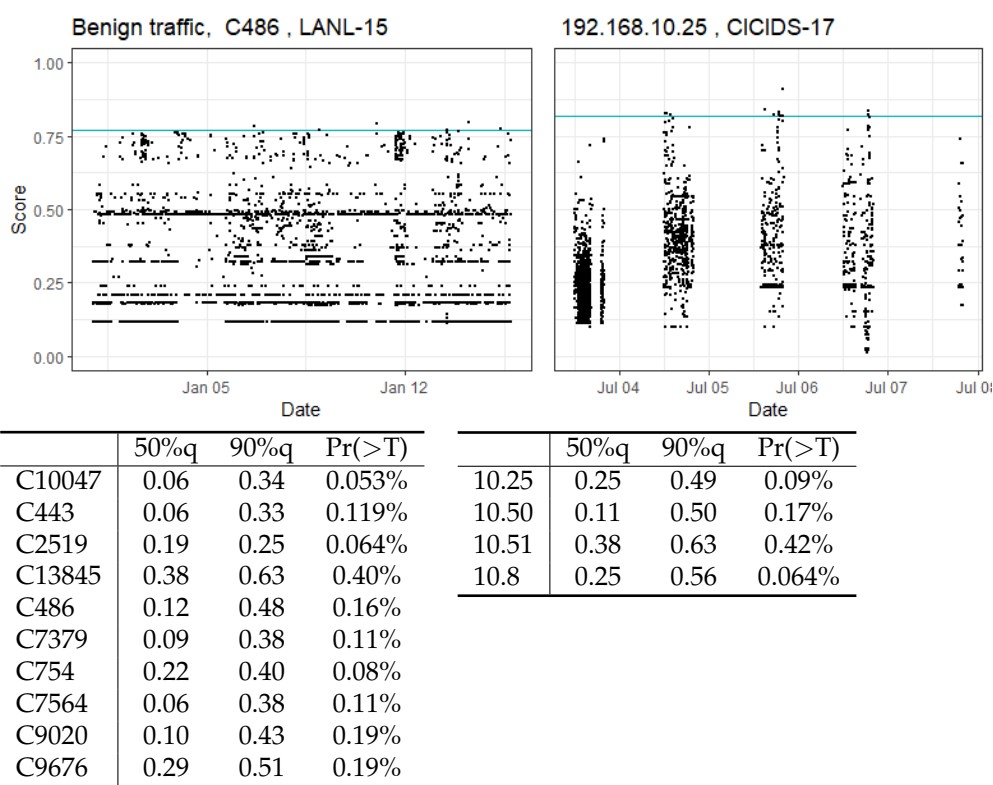

|         | 50%q | 90%q | Pr(>T)  |       | 50%q | 90%q | Pr(>T)  |
|---------|------|------|---------|-------|------|------|---------|
| C10047  | 0.06 | 0.34 | 0.053%  | 10.25 | 0.25 | 0.49 | 0.09%   |
| C443    | 0.06 | 0.33 | 0.119%  | 10.50 | 0.11 | 0.50 | 0.17%   |
| C2519   | 0.19 | 0.25 | 0.064%  | 10.51 | 0.38 | 0.63 | 0.42%   |
| C13845  | 0.38 | 0.63 | 0.40%   | 10.8  | 0.25 | 0.56 | 0.064%  |
| C486    | 0.12 | 0.48 | 0.16%   |       |      |      |         |
| C7379   | 0.09 | 0.38 | 0.11%   |       |      |      |         |
| C754    | 0.22 | 0.40 | 0.08%   |       |      |      |         |
| C7564   | 0.06 | 0.38 | 0.11%   |       |      |      |         |
| C9020   | 0.10 | 0.43 | 0.19%   |       |      |      |         |
| C9676   | 0.29 | 0.51 | 0.19%   |       |      |      |         |

**Figure 10.** Anomaly score distributions for benign traffic in the LANL-15 and CICIDS-17 datasets.

### 6.3. Importance of Training Data Size

Host C13845 in the LANL-15 and host 192.168.10.51 in the CICIDS-17 data are exceptions among the above observations, with their median anomaly score each being 0.38 and their estimated false-positive rates being 0.4% and 0.42%, which significantly exceeds the average of 0.1%.

When examining host 192.168.10.51, we noticed that it produced less traffic than other hosts in the CICIDS-17 data. Due to this fact, the training dataset only contains 3096 sessions or 36,989 flows for this host, compared to about 10,000 sessions or 115,000 for host 192.168.10.25.

For host C13845, we observe a similar picture. Since the host is less active than others in the dataset, the training data only contain 728 sessions or 2423 flows for this host, compared to 6013 sessions for the host with the next fewest training sessions.

This suggests that traffic on these hosts is not necessarily harder to predict for CBAM, but that the lack of sufficient training data prevents CBAM from learning traffic patterns for these two hosts effectively. To verify this, we examined how many sessions are necessary

in the training phase to achieve similar false positives at a given anomaly threshold. We selected the hosts with the most sessions for each dataset and reduced the number of training sessions from 10,000 to 3000 and 1000. We then trained models with otherwise similar settings and compared how many additional sessions exceeded the anomaly threshold. For UGR-16 and LANL-15, we examined whether we could increase the number of training sessions to 20,000, which was not possible for the CICIDS-17 data.

Figure 11 depicts the corresponding false-positive rates for each host. False-positive rates for the UGR-16 and the CICIDS-17 hosts are already significantly affected when only trained on 3000 sessions and increase further when only 1000 host sessions are available during training. Increasing the number of sessions to 20,000, however, does not seem to have an effect to further improve the model.

For the host in the LANL-data, this effect is far less pronounced, and false-positives at 3000 sessions are similar to the ones at 10,000 and 20,000. CBAM is apparently able to learn flow predictions sufficiently from similar hosts in the dataset without depending on sessions specifically from the selected host. When we train CBAM exclusively on sessions from host C2519 without data from other hosts, the same deterioration of model prediction can be observed.

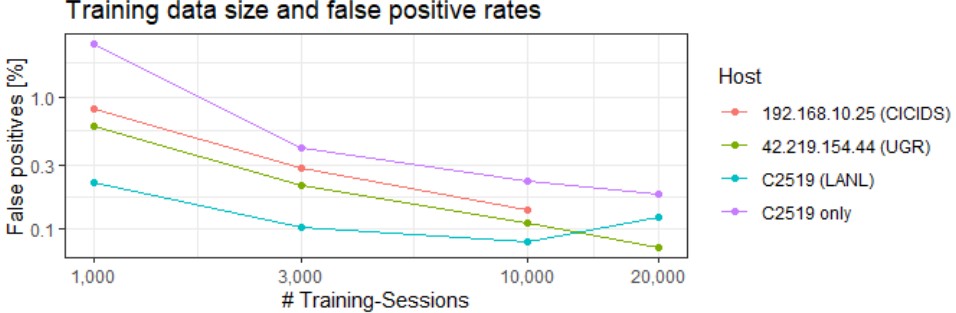

**Figure 11.** Influence of number of sessions in training data for benign traffic modelling accuracy.

## 7. Traffic Analysis

### 7.1. How Attacks Affect Flow Structures

We now examine in more detail why modelling sequences of flows is effective to detect access attacks, and how these attacks alter common flow structures. Unfortunately, the CICIDS-17 dataset—and to our knowledge, all other NID-datasets—do not contain sufficient ground truth information about included attack traffic, so this analysis is based on empirical domain-knowledge of similar attacks as well as the traffic itself.

Figure 12 shows a session in the CICIDS-17 data that corresponds to a SQL-injection attack on host 172.16.0.1, a Ubuntu web server. Depicted below is the order of the flows along with their direction, the destination port and the size of the flow. Dashed rectangles indicate the most likely flow size as predicted by CBAM. Below are the likelihoods of the actually observed flow sizes on a log-scale, which determine the anomaly-score of the session.

SQL requests from a web-server typically consist of the verification of user credentials or the retrieval of specific content on a webpage. In an SQL-injection, SQL-code is injected into a HTTP-request that forces the server to retrieve, modify or forward additional content from an SQL-database, which can significantly increase the size of the corresponding SQL-request.

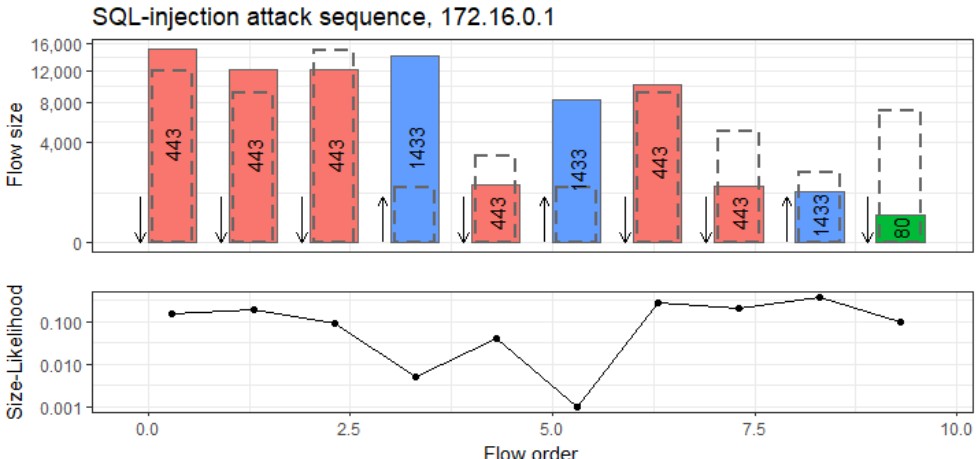

**Figure 12.** Flow-sequence in an SQL-injection attack with predicted size likelihood (log-scale). Arrows indicate flow directions (down = incoming; up = outgoing).

The sequence of flows in Figure 12 overall resembles regular incoming HTTP requests accompanied by corresponding outgoing SQL-requests from the server. However, Figure 12 clearly shows that the sizes of two of the SQL-connections on port 1433 are magnitudes larger than predicted by CBAM based on the context of the surrounding flows, which is likely caused by the injection attack. This results in a very low likelihood of the observed flow sizes and a high anomaly score for the whole session.

Figure 13 depicts a session that corresponds to an infiltration attack on host 192.168.10.8 in the CICIDS-17 data. Again, the figure depicts the order, direction, size and destination port of the flows, along with predictions of the most-likely sizes (dashed rectangles) and the overall likelihood of the actually observed flows.

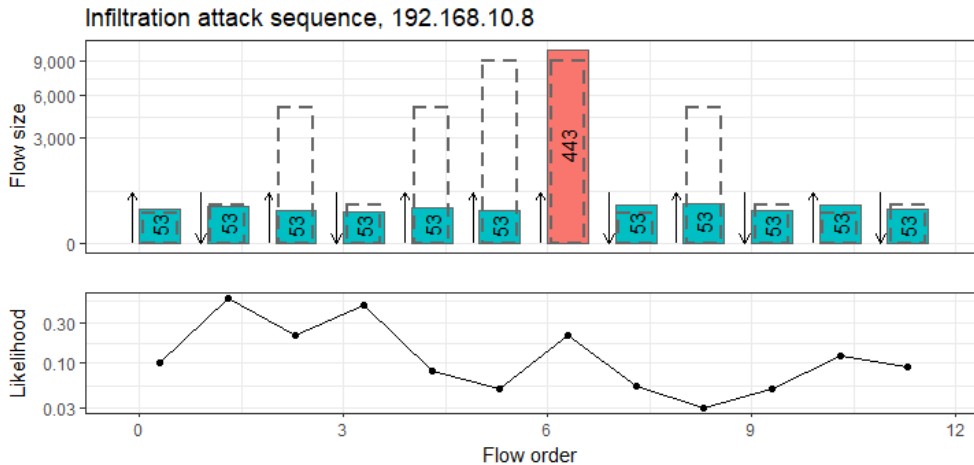

**Figure 13.** Flow-sequence in an infiltration attack with port-direction likelihood (log-scale).

This sequence does not resemble the regular behaviour typically encountered on this host. DNS flows on port 53 are typically followed by HTTP flows on port 80 or 443, to which the model assigns a very high likelihood after the first four flows. However, this session contains many excessive consecutive DNS flows, which are interrupted by only one HTTP flow. Correspondingly, the likelihood for the excessive DNS flows as well as the overall session likelihood is low.

It is not completely clear how the infiltration attack triggers this abnormal behaviour. Possibly, the infiltration software is trying to retrieve the current address of a C&C-server via DNS.

### 7.2. Benign Traffic Bands and Cause of False-Positives

Figure 9 displays how benign traffic is clustered into stable bands. We coloured three of these bands at different levels as well as two of the observed false-positives, which we are now examining closer. Figure 14 depicts the corresponding dominant session pattern that is present in each band along with the predicted likelihood for each flow. Again, the figure depicts the order, direction, size and destination port of the flows, along the overall likelihood of observed flows. For clarity, we omitted the predictions of the most-likely flow sizes (dashed rectangles).

The two lower bands, blue and red located at $AS = 0.061$ and $AS = 0.18$, represent simple and frequent HTTP- as well as corresponding NoSQL-requests and SSH activity by the server. These sessions are therefore predicted with high accuracy.

The green band at $AS = 0.45$ contains more complex and longer sessions that involve both incoming and outgoing HTTP-connections as well as TelNet and RDP connections. The size and order of the flows in these sessions is less deterministic than the activity in the red and blue bands. This activity is also less frequent, which explains the less accurate predictions by CBAM. The model, however, still recognises these sessions and is able to predict flow state and size with a non-vanishing probability, which keeps the overall session score bounded.

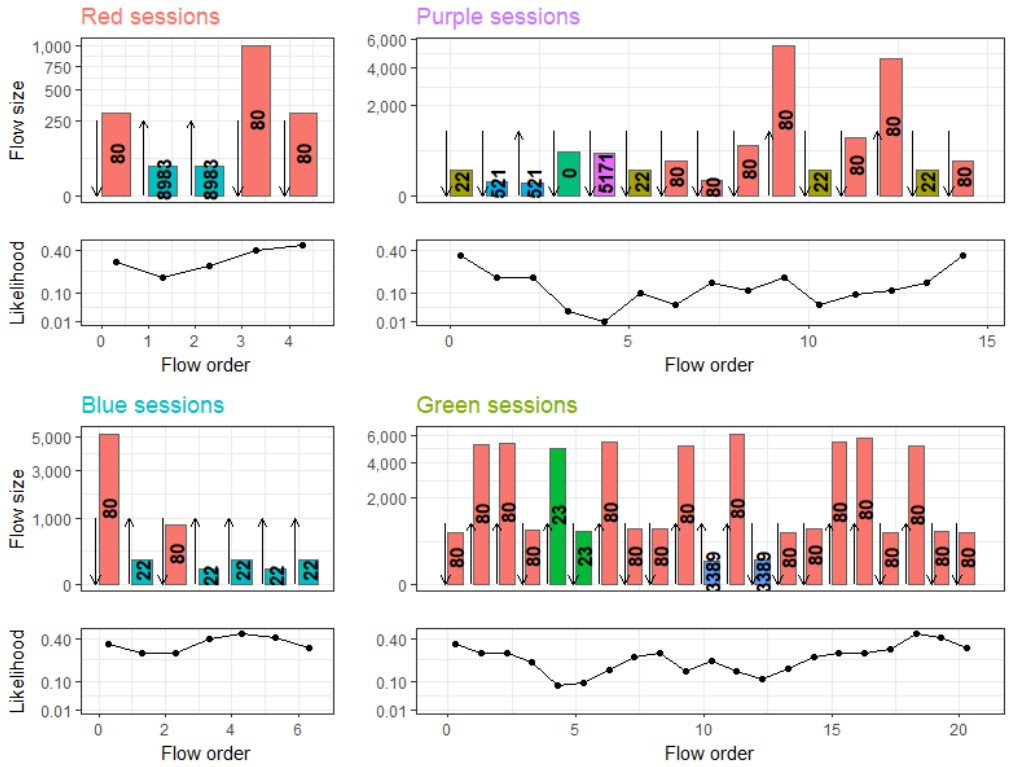

**Figure 14.** Sessions corresponding to score banding structures in Figure 9, with predicted likelihoods (log-scale).

The two purple-coloured sessions likely represent server inspection activity, involving activity on port 0, SSH-sessions and activity on uncommon ports. This type of activity is very rare on this server and appears less deterministic than other more common activity. CBAM therefore fails to recognise the session structure and is not able to assign non-vanishing probabilities to several flows, which decreases the overall session likelihood and results in a high anomaly-score. We are not aware of how often servers are subject to inspections and whether this would present a problem in operational deployment. However, it seems feasible that resulting false-alerts could be linked to this administrative activity automatically or by a security analyst.

## 8. Benefit of Specific Design Steps

A significant part of the conducted work was concerned with improving the given network design to address insufficient predictions for several traffic phenomena and boost overall model performance. We now outline several key-steps in the design process and how they improve performance.

### 8.1. Bidirectionality for Better Session Context

The usage of bidirectional LSTM layers compared to unidirectional ones significantly improved the prediction of events at the beginning of a session and consequently boosted the detection rates within short sessions. Figure 15 demonstrates this in a detailed manner: displayed is a short session of four flows containing FTP and HTTP activity on host "42.219.153.32". On the right side are the predicted likelihoods of FTP and HTTP states for each flow in the session, with the blue bars corresponding to the predictions by the forward layer, while the red bars display the backwards direction and the green bars display the likelihoods after aggregating both predictions.

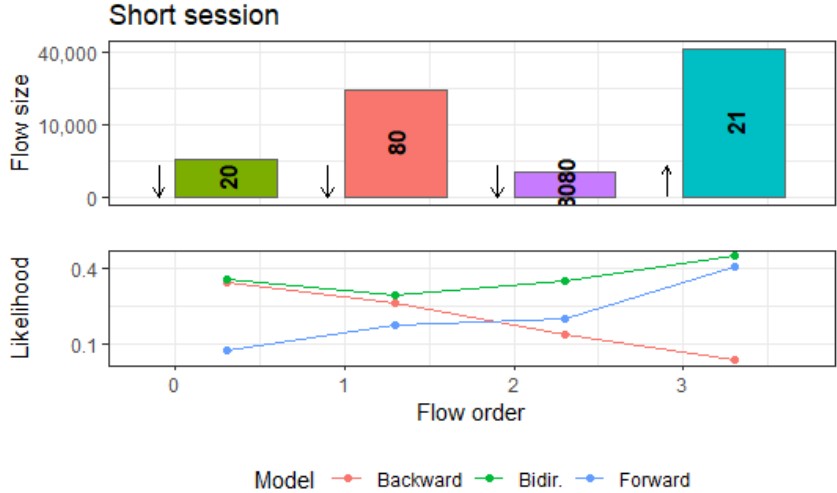

**Figure 15.** Common short session and the flow likelihoods predicted by each dirctional model.

Figure 15 demonstrates this in a detailed manner: Displayed is a short session of four flows containing FTP and HTTP activity on host "42.219.153.32". On the right side are the predicted likelihoods of FTP and HTTP states for each flow in the session, with the blue bars corresponding to predictions by the forward layer, while the red bars display the backwards direction and the green bars display the likelihoods after aggregating both predictions.

When only relying on the forward direction, for the first two flows, the predicted likelihoods are less than 0.07 each. The last two flows of the session are, however, well predicted with high likelihoods over 0.3. Because the session is short, the inaccurate predictions for the first two flows decrease the overall likelihood of the session to 0.18 and the corresponding anomaly score to $AS = 0.73$, which is just below the anomaly threshold, even though this type of flow sequence is quite common in the UGR-16-dataset. In a similar manner, this applies to the backward direction with the likelihood of the last two observed flows being 0.02 and 0.03, respectively.

The cause for these phenomena is that the start of a session can differ significantly for different activities, and the LSTM-layer needs some context before recognising the specific activity and make corresponding predictions. In short sessions, the lack of accurate predictions in the first flows can then dominate the anomaly-score of the whole session.

By adding a bidirectional layer, we are able to provide context for these initial flows in a session as well by looking at later flows first. The green bars in Figure 15 displays this: by basing predictions both on the output of the forward- and the backward-layer, the

bidirectional model is able to predict flow likelihoods significantly better and thus assign the session a much lower anomaly score.

Table 6 displays how much we could decrease false-positive rates by replacing the unidirectional LSTM layer with a bidirectional one. Overall, the false-positives decreased by 61% for the CICIDS-17 dataset, and by 52% for the UGR-16 dataset. More strikingly, when only looking at short sessions that contain less than five flows, we were able to reduce false-positives by 94% and 87%, respectively.

**Table 6.** Average likelihood of first two flows in a session and false-positives for uni- and bidirectional model.

| | | Likelihood of Flows 1 and 2 | | FP Rate [%] | |
|---|---|---|---|---|---|
| | | **Unidir** | **Bidir** | **Unidir** | **Bidir** |
| UGR-16 | All sess. | 0.13 | 0.27 | 0.31 | 0.12 |
| | sess. < 5 flows | 0.19 | 0.41 | 1.6 | 0.09 |
| CICIDS-17 | All sess. | 0.09 | 0.29 | 0.37 | 0.18 |
| | sess. < 5 flows | 0.05 | 0.30 | 1.7 | 0.13 |

### 8.2. Additional Layers for Complex Session Modelling

The inclusion of a second LSTM-layer as well a subsequent linear layer allows CBAM to capture more complex behaviour in long sessions as well as remember rare behaviour more quickly. It also increased the average predicted likelihood for flows overall.

To examine the benefit of the described model depth, we compare it to a more shallow version that lacks the second LSTM- and linear layer, as depicted in Figure 16, which was trained under otherwise similar conditions. Here, we examine in detail how the increased model depth allows better predictions for complex sessions.

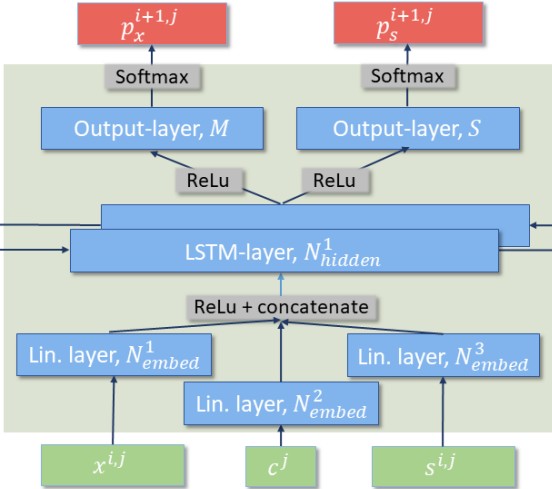

**Figure 16.** Architecture of the shallow LSTM model.

Figure 17 displays two different types of activities, A and B, which are common in the CICIDS-17 data. The structure in these sessions can be frequently observed with only minor variations. Consequently, the sessions are predicted well by both the original and the more shallow model.

However, traffic from two or more activities can sometimes occur simultaneously and thus become grouped into the same session. Figure 17 shows how the traffic from activities A and B are overlapping in a session, which makes the structure in the session more complex to predict.

The displayed likelihoods show that predictions by the shallow model are accurate for flows at the beginning of the session, but these deteriorate once they encounter flows from activity B. Prediction accuracy by the more complex model is also decreasing but remains on a sufficient level to assign this session an anomaly score of $AS = 0.51$, compared to $AS_S = 0.79$ for the shallow model. When looking at the activation in the LSTM-memory cell, we see that similar neurons as in activity A are activated at the beginning of the session, which shifts during the course of the session and resembles a more similar activation as in activity B at the end.

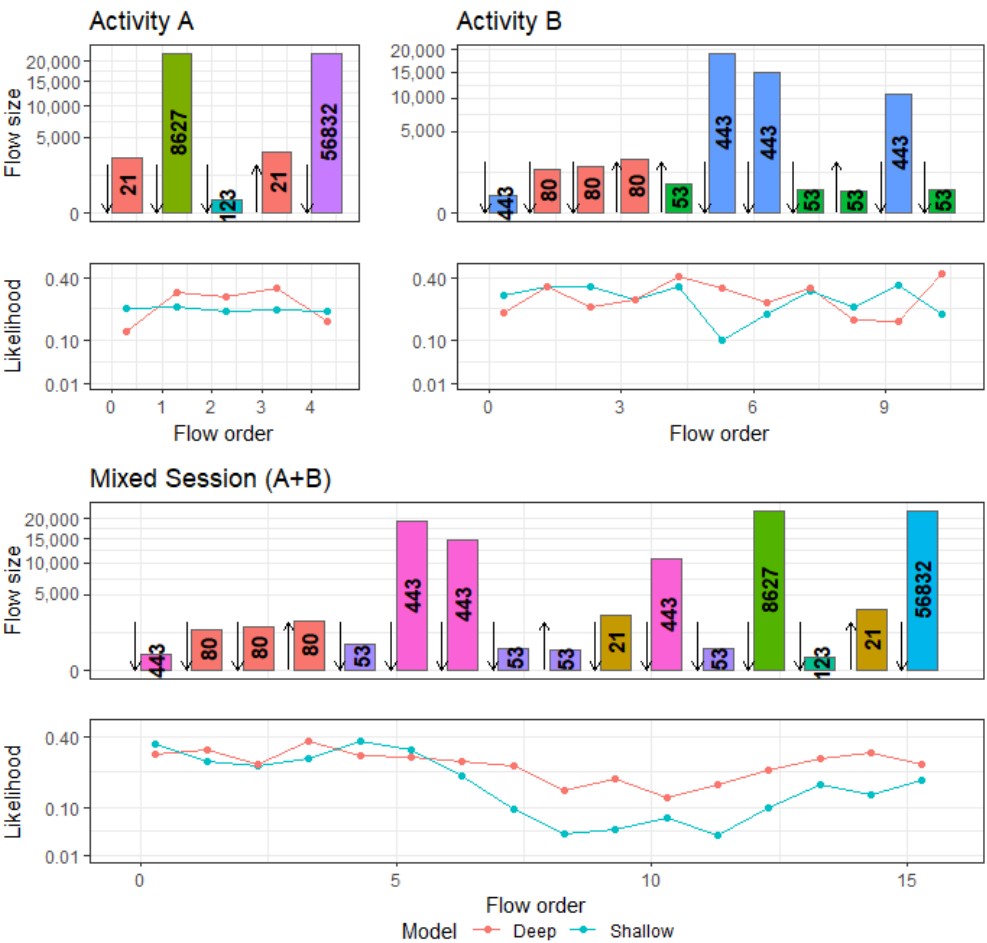

**Figure 17.** Predictions for two activities in isolated sessions and in a mixed session.

Overall detection rates for this model can be found in Table 4, while score distributions can be found further down in Table 7.

**Table 7.** Score distributions for simpler models.

|  | Shallow LSTM | Markov MC | NDFA |
|---|---|---|---|
| Ben. 50%q | 0.22 | 0.61 | 0.55 |
| Ben. 90%q | 0.55 | 0.81 | 0.86 |
| Ben. 99%q | 0.73 | 0.89 | 0.96 |
| Mal. 50%q | 0.70 | 0.60 | 0.68 |
| Mal. 90%q | 0.85 | 0.83 | 0.89 |
| Mean AUC | 0.86 | 0.53 | 0.64 |

The improvements achieved by adding these additional layers could suggest that increasing the number of layers even further will decrease false positive rates even further, which we discuss in Section 10.

*8.3. Comparison with Simpler Models*

In this section, we aim at studying whether the higher complexity of an LSTM network is necessary for the task of detecting contextual network anomalies, or whether simpler baseline methods can achieve the same results. For comparison purposes, we implemented a first-order Markov chain (MC) and a non-deterministic finite automata (NDFA) model. Both methods are widely used in sequence modelling, and have been successfully applied to security problems [30,31]. In contrast to LSTMs, Markov chains have no memory past the last event while NDFAs can distinguish between different types of sequences via state-merging, and give corresponding transition probabilities.

Similar to our LSTM model, Markov Chains and Finite Automata predict state transition probabilities, which is why we can employ the same anomaly score computation. However, computational costs increase quadratically with the number of states, and a separation of state vocabularies is not possible. We restrict both comparison models to the above-described port-direction states. Models and detection rates were determined on the CICIDS-17 dataset.

Table 7 shows distribution characteristics of benign and malicious sessions for our shallow LSTM-model, the Markov chain model and the NDFA. It shows that CBAM outperforms these baseline methods, but also that the automata performs better than Markov chains. While the Markov chain is practically not able to make any distinction between malicious and benign traffic, the automata model shows some, albeit limited, ability to identify anomalous sessions, mainly for the three types of brute-force attacks. This order shows the importance of sequence memory for contextual anomaly detection, and confirms our previous comparison of the suitability of Markov chains and NDFAs for network intrusion detection [32].

## 9. Related Work

The application of recurrent neural networks to network intrusion detection has risen in popularity recently. LSTM models for web attack detection, such as by Yu et al. [33], improve the detection rates of simpler preceding models such as Song et al. [34]. They rely on deep packet inspection and are often targeted at protecting selected web-servers rather than network-wide, due to a lack of computational scalability and increasing traffic encryption. Methodologically, vocabularies are created from string sequences with well-known NLP methods, while CBAM provides a new vocabulary-construction method suitable for traffic metadata.

The majority of LSTM-based metadata approaches rely on labelled attack data for classification, and do not have the scope of anomaly-based models to detect previously unseen attacks. A prominent example of this comes from Kim et al. [6], who classify flow sequences based on 41 numeric input features. Anomaly-based approaches such as ours mostly rely on iterative one-step-ahead forecasts, with the forecasting error acting as the anomaly indicator. This was for instance done in GAMPAL by Wakui et al. [35], who used flow data aggregation as numerical input features, which are computationally easier to process, but cannot encapsulate high-level information such as the used protocol, port, or direction. These models are best used for detecting high-volume attacks. Apart from our work, only Radford et al. [7] created event vocabularies from flow protocols and sizes. We use a more sophisticated model in terms of stacked recurrent layers and embeddings for more input features, which results in higher detection rates, as demonstrated in see Section 8. The HCRNNIDS model by Kahn provides an interesting adaption of hybrid convolutional recurrent networks typically used in video modelling for intrusion detection [36] with promising results. In comparison to CBAM, this model is applied to individual flow features rather than flow sequences, and is trained as a classifier rather than an anomaly-detection model.

Encoder–decoder models are increasingly used in combination with LSTM networks to create the embeddings of packet or flow sequences, such as that done by Zhong et al. [37] for anomaly detection. Zhou et al. [38] used embeddings to facilitate anomaly-detection that is robust against dataset imbalances. Liu et al. [29] use embeddings to augment and inflate minority class data samples for the same purpose.

Berman et al. [39] surveyed recent deep-learning techniques for network intrusion detection as well as other cyber-security applications. They assess whether recurrent methods are currently state-of-the-art, but do not reach a conclusion as to whether they perform better than convolutional or generative methods.

Notable work outside of network traffic includes Tiresias [40], an LSTM model for security event forecasting with great accuracy, and DeepLog [41], an LSTM network to learn a system's log patterns (e.g., log key patterns and parameter values) from normal execution. The design of Tiresias has similarities to ours, but the scope of the model is attack forecasting rather than intrusion detection and relies on both different input data in the form of IDS logs as well as different evaluation metrics. DeepLog is combined with a novel log parser to create a sequence of symbolic log keys, which is then also modelled using one-step forecasting. The authors achieve good detection results in regulated environments such as Hadoop with a limited variety of events (e.g., 29 events in Hadoop). Here, CBAM goes further by being applied to a much more heterogeneous data source and creating a more than 30 times larger vocabulary. Han et al. [42] recently proposed *UNICORN*, a deep graph-net-based anomaly-detection method for provenance-based data that demonstrates how effective neural anomaly-detection methods are at detecting unknown intrusions.

## 10. Limitations and Evasion

### 10.1. Limitations

CBAM is an initial application of short-term contextual modelling on network traffic that demonstrates the potential of contextual traffic models for intrusion detection. Although we used a relatively simple model with few but carefully selected input features, we outperformed sophisticated methods while retaining low false positive rates. The detection rates are to be taken with care as the available access attack data are small, synthetic and contain only a limited number of attack classes. The detection rates in the cross-evaluation on a real-world access attack in the LANL-15 data gives us confidence that CBAM's performance is reproducible in real-world scenarios, but additional data are required for an ultimate conclusion.

A frequently asked question concerns whether low false-positive rates carry over from the synthetic background traffic in datasets such as CICIDS-17 to real-world scenarios [43]. We believe that this was sufficiently demonstrated in the long-term evaluation and the observed score stability on the UGR-16 real-world dataset.

The improvements achieved by adding additional layers could suggest that increasing the number of layers will even further decrease false positive rates, which is certainly worth exploring in future work. However, as discussed in Section 6.1, the current main source for false positives are rare activity events which are not contained in the training data and are therefore not be recognised by the model. To make a significant reduction in the false positive rate, we would need to train on datasets spanning more computers or over longer time periods. We are, however, aware of the difficulties involved in creating datasets for NIDS evaluation.

### 10.2. Evasion and Resilience

Evasion tactics and corresponding model resilience against them have been a concern in the development of NIDS. We specifically focused on short-term sequential anomalies as they are often an unavoidable by-product of attack sequences, and it is thus very difficult for an attacker to perturb attack sequences that rely on a specific exploit without pre-existing control over the victim device or other network devices. We therefore believe that CBAM

is relatively robust against evasion. However, we identified potential improvements for future work.

A specific evasion tactic that has been discussed extensively in the context of machine learning is model poisoning in the training/retraining phase. A great difficulty for the attacker is the fact that CBAM uses sequences of symbolic events rather than continuous parameters. The introduction of a gradual shift is therefore not possible in a direct way as the alteration of individual events would look anomalous straight away. Furthermore, it is normally not possible for an attacker to alter individual events significantly without pre-existing control over network devices or specific exploits, i.e., the change of the port or size would normally cause an error in the communication. It is thinkable that an attacker could increase the predicted probability of specific events patterns more gradually by overlaying traffic stemming from third party devices. However, the attacker would either need control of these devices or the ability to monitor traffic to the victim device in real-time, neither of which are usually available. We also showed in Section 6 that short-term contextual traffic patterns remain stable over several months, which means that retraining CBAM is only necessary at a low rate and attackers will have to wait for a long time to execute successful model poisoning.

An issue we encountered is the overlay of malicious and benign traffic. Currently, the existence of known traffic patterns in a session can deplete the overall anomaly score of a session. A potential evasion tactic could therefore try to conceal an attack behind benign communication on the victim device, an already common approach for C&C communication. Possible improvements for this issue are a refined notion of a session that groups related traffic better, and a better scoring method that identifies smaller anomalous sequences in an otherwise normal sequence of flows. Additionally, developing more sophisticated detection methods from the computed scores may boost detection rates.

## 11. Conclusions

CBAM presents a new and promising angle to anomaly-based intrusion detection that significantly improves detection rates on the types of network attacks with the lowest detection rates. We use an anomaly-based approach that does not rely on specific notions of attack behaviours and is therefore better suited at detecting unknown attacks rather than regular misuse- or signature-based systems. By assigning contextual probabilities to network events, CBAM improves the detection rates of low-volume remote access attacks and outperforms current state-of-the-art anomaly-based models in the detection of several attacks while retaining significantly lower false positive rates. Furthermore, CBAM retains low false positive rates for periods stretching several months. Our results provide good evidence that using contextual anomaly detection may in the future help decrease the threat of previously unseen vulnerabilities and malware aimed at acquiring unauthorised access on a host. We specifically focused on short-term anomalies as they are often an unavoidable by-product of an attack and thus very difficult for an attacker to avoid without pre-existing control over the victim device or other network devices.

**Author Contributions:** Conceptualisation, G.G.; formal analysis, H.C.; methodology, H.C., G.G. and D.A.; supervision, D.A.; validation, H.C.; visualisation, H.C.; writing—original draft, H.C.; writing—review and editing, G.G. All authors have read and agreed to the published version of the manuscript.

**Funding:** The authors gratefully acknowledge funding support from BT Group PLC, UKRI/EPSRC (grants EP/N510129/1 and EP/L02277X/1), ONR Global (award N62909-17-1-2065) and the hosting University of Edinburgh departments, EPCC and School of Informatics.

**Data Availability Statement:** The datasets analysed in this work are publicly available and can be found as follows: CICIDS-17 dataset [14], https://www.unb.ca/cic/datasets/ids-2017.html (accessed on 10 June 2021), UGR-16 dataset [16] https://nesg.ugr.es/nesg-ugr16/ (accessed on 10 June 2021), LANL-15 dataset [15], https://csr.lanl.gov/data/cyber1/ (accessed on 10 June 2021).

**Conflicts of Interest:** The authors declare no conflict of interest.

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
