# Peer review of "CBAM: A Contextual Model for Network Anomaly Detection"

_computers, doi:10.3390/computers10060079_

Round 1

Reviewer 1 Report

The work entitled “Flows in a back-and-forth context: Detecting access attacks with bidirectional anomaly models” presents a method to improve the network intrusion detection based on the LSTM deep-learning technique.

In particular, the authors focus on a bi-directional approach of LSTM useful to improve the intrusion detection and examine other aspects such as the impact of the training set size in recognising benign traffic.

The experimental analysis has been carried out by considering some real-world datasets such as CICIDS-17, LANL-15, UGR-16.

The work seems interesting and sounding. I have just a couple of minor concerns.

1. Some notation must be simplified or better explained. For example, in p_x^{i,j,k} and  p_s^{i,j,k} the authors must explain better the conditional term {x,s}^{1:i-1,j}.

2.The Related work section should be improved, by considering recent works which deal with affine themes, and in particular, the network intrusion detection assisted by Deep-based techniques. Some suggestions follow:

- Intrusion Detection of Imbalanced Network Traffic Based on Machine Learning and Deep Learning (IEEE Access, 2021), where techniques such as LSTM, Random Forest, SVM, XGBoost are compared among them, along with the proposal of an algorithm to tackle the class imbalance problem.

- Experimental Review of Neural-based Approaches for Network Intrusion Management (IEEE Transactions on Network and Service Management, 2020), where some neural-based techniques (including LSTM) are evaluated and compared with other techniques on datasets such as the CICIDS-17, and where the impact of the dataset size is investigated.

- Implementing a Deep Learning Model for Intrusion Detection on Apache Spark Platform (IEEE Access, 2020), where deep learning methods for intrusion detection systems are implemented through Apache Spark platform.

3. Some graphical adjustments are needed

- y-axis in Fig. 13 is not indicated (Score probably)

- Fig. 5, “ICMP” label (on x-axis) is cutted

Reviewer 2 Report

Anomaly-based intrusion detection is an increasingly widely used addition to signature-based methods to provide protection from previously unseen attacks in Network Intrusion Detection Systems (NIDS). Most current anomaly-based NIDS use aggregated traffic features and succeed well in detecting group anomalies such as network probes or worms. Recent evaluations show that the current anomaly-based network intrusion detection methods fail to detect remote access attacks reliably [1]. These are smaller in volume and often only stand out when compared to their contextual surroundings, which we call contextual anomalies. We present and examine a deep bidirectional LSTM approach that is designed specifically to detect such attacks as contextual network anomalies. The paper is interesting overall, but following are the comments that must be addressed:

 Comments:

  • In my opinion, the Title of the paper is not suitable it looks lengthy and a Good paper has a short Title so authors should think about it.
  • Authors need to re-write the Abstract in a more meaningful way example (Problem definition=> How existing methods are lacking => proposed solution => Outcome).
  • The proposed solution is very briefly described. One can understand that is enough to obtain your results by couple bi-LSTM. A lot of information should be provided here about the considered/developed architecture.
  • How do you deal with the network overfitting problem?
  • The authors need to explain How do you deal with the dataset imbalance problem? Did you consider any data augmentation techniques? Is any time complexity issue in brute force algorithm?? How you handle this issue ?? don’t discuss it anywhere.
  • Authors should Use a methodology described in Ulvila, Gaffney, “Evaluation of Intrusion Detection Systems” to evaluate the proposed intrusion detection approach. Present and discuss the False alarm probability vs. Probability of detection plot.
  • Authors should consider adding a short discussion on the recent available public network intrusion datasets (such as Boğaziçi University, CSE-CIC-DS2018 data, and LITNET-2020 dataset) and summarizing their advantages and limitations.

Erhan, Derya, and Emin Anarım. "Boğaziçi University distributed denial of service dataset." Data in brief 32 (2020): 106187. Khan MA. HCRNNIDS: Hybrid Convolutional Recurrent Neural Network-Based Network Intrusion Detection System. Processes. 2021; 9(5):834. https://doi.org/10.3390/pr9050834. Damasevicius, R., Venckauskas, A., Grigaliunas, S., Toldinas, J., Morkevicius, N., Aleliunas, T., & Smuikys, P. (2020). LITNET-2020: An annotated real-world network flow dataset for network intrusion detection. Electronics, 9(5), 800.

  • Before Conclusion, please Discuss the limitations of the proposed approach and the threats to the validity of the experimental results.

Round 2

Reviewer 2 Report

Authors solved try previous comments, but this paper still need to improve

The structure of the paper is very confusing for the readers so authors should make it simple and interesting for the readers.

The authors still missing a few experiment paraments so authors should provide all parameters for the readers.

Authors need to justify section 4.3 “This is however not necessary in our case, as training for anomaly-detection is done independently of the minority class”. ID results and performance without data imbalance is not any contribution and results are biased so authors should elaborate and justify it more technically for readers.

Authors should include these references and even in the entire paper authors did not include the latest paper of 2021.

Erhan, Derya, and Emin Anarım. "Boğaziçi University distributed denial of service dataset." Data in brief 32 (2020): 106187. Khan MA. HCRNNIDS: Hybrid Convolutional Recurrent Neural Network-Based Network Intrusion Detection System. Processes. 2021; 9(5):834. https://doi.org/10.3390/pr9050834. Damasevicius, R., Venckauskas, A., Grigaliunas, S., Toldinas, J., Morkevicius, N., Aleliunas, T., & Smuikys, P. (2020). LITNET-2020: An annotated real-world network flow dataset for network intrusion detection. Electronics, 9(5), 800

Why authors select these three ID datasets(CICIDS-17, LANL-15, UGR-16) while several ID datasets are available for any specific reason??

Section 9.1 Related work at end of paper ??/

There are typo and formating issues authors should need to improve the quality of the paper.

Round 3

Reviewer 2 Report

The authors did excellent work and the paper looks very inserting for readers so now this paper is accepted for publication in the current form.